# Application of Pressurized Liquid Extractions to Obtain Bioactive Compounds from *Tuber aestivum* and *Terfezia claveryi*

**DOI:** 10.3390/foods11030298

**Published:** 2022-01-23

**Authors:** Eva Tejedor-Calvo, Sergi García-Barreda, Sergio Sánchez, Asunción Morte, María de las Nieves Siles-Sánchez, Cristina Soler-Rivas, Susana Santoyo, Pedro Marco

**Affiliations:** 1Centro de Investigación y Tecnología Agroalimentaria de Aragón (CITA), Instituto Agroalimentario de Aragón—IA2 (CITA-Universidad de Zaragoza), Avda. Montañana 930, 50059 Zaragoza, Spain; sgarciaba@cita-aragon.es (S.G.-B.); ssanchezd@cita-aragon.es (S.S.); pmarcomo@cita-aragon.es (P.M.); 2Department of Production and Characterization of Novel Foods, Institute of Food Science Research—CIAL (UAM + CSIC), C/Nicolas Cabrera 9, Campus de Cantoblanco, Universidad Autónoma de Madrid, 28049 Madrid, Spain; maria.siles@uam.es (M.d.l.N.S.-S.); cristina.soler@uam.es (C.S.-R.); susana.santoyo@uam.es (S.S.); 3Departamento de Biología Vegetal, Facultad de Biología, Campus de Espinardo, Universidad de Murcia, 30100 Murcia, Spain; amorte@um.es

**Keywords:** *Tuber aestivum*, *Terfezia claveryi*, response surface methodology, immunomodulation, antioxidants, enzymes

## Abstract

A PLE (pressurized liquid extraction) method was adjusted following a full-factorial experimental design to obtain bioactive-enriched fractions from *Tuber aestivum* and *Terfezia claveryi*. Temperature, time and solvent (water, ethanol and ethanol–water 1:1) parameters were investigated. The response variables investigated were: obtained yield and the levels of total carbohydrate (compounds, β-glucans, chitin, proteins, phenolic compounds and sterols). Principal component analysis indicated water solvent and high temperatures as more adequate parameters to extract polysaccharide-rich fractions (up to 68% of content), whereas ethanol was more suitable to extract fungal sterols (up to 12.5% of content). The fractions obtained at optimal conditions (16.7 MPa, 180 °C, 30 min) were able to protect Caco2 cells from free radical exposure, acting as antioxidants, and were able to reduce secretion of pro-inflammatory cytokines in vitro: IL-6 (50%), and TNFα (80% only *T. claveryi* ethanol extract), as well as reduce high inhibitory activity (*T. aestivum* IC_50_: 9.44 mG/mL).

## 1. Introduction

Truffles are the hypogeous fruiting bodies of fungi that form mycorrhizal associations with particular tree or shrub species. Taxonomically, edible truffles belong to the *Tuberaceae* family and the *Tuber* genus, including most of the species appreciated mainly by organoleptic properties. Besides, other important truffle genera (*Terfezia* and *Tirmania*) with culinary and medicinal interests are named desert truffles [1,2].

The main nutritional constituents of truffles are carbohydrates, followed by proteins [3,4,5]. Most of their carbohydrates are considered dietary fibers such as chitin, β-glucans and other polysaccharides, and they also include mannitol and trehalose [6] as well as smaller sugars such as D-glucose, D-mannose or D-galactose [7]. Although truffles show low fat levels, their lipid content is important since they are involved in flavor and aroma properties. To maintain their hyphal membranes it is necessary to obtain unsaponificable molecules such as ergosterol (ergosta-5,7,22-trienol), ergosta7,22-dienol, stigmasterol or ergosta-5,8-dieno-3-ol [8,9]. Brassicasterol (ergosta-5,22-dienol) is also frequently detected in truffles; however, it is mainly reported in plants and algae species but is also found in species belonging to the subphylum *Taphrinomycotina*, a dimorphic plant parasite [10]. Thus, it might also be present in the fungus because of its close interactions with their host plant [11]. These fungal compounds were previously pointed to as molecules with interesting biological activities promoting human health beyond simple nutrition. According to several reports, fungal β-glucans showed immunomodulatory and hypocholesterolemic properties, among others [12,13]. Fungal sterol-enriched fractions also reported some bioactivity such as hypocholesterolemic [14,15]. Recently, truffles have shown interesting bioactive compounds, and their potential bioactivities are now being studied, e.g., antitumoral, antioxidant, immunomodulatory and hypoglucemic properties [7,16,17,18,19].

Among the environmentally friendly extraction technologies, PLE (pressurized liquid extractions) was successfully adjusted to obtain a wide range of bioactive fractions from yeast and mushrooms to design novel functional foods [3,20,21]. In comparison with conventional extraction techniques (soxlhet, maceration, solvent extraction), PLE involves some advantages such as: time saving, less solvent consumption, and use of nontoxic solvents (water or ethanol), among others [22]. However, studies to optimize PLE methods to extract bioactive ingredients from truffles are scarce, particularly from those of desert truffles [7,23]. Truffles are usually consumed fresh and during special occasions; however, with the improvements implemented during their cultivation, truffle yields are exponentially increasing and becoming more and more accessible. The low-quality specimens (ugly shape, small size, broken pieces, etc.) might be valorized as functional ingredients, as they share many bioactive compounds with mushrooms, and they might also contain other particular compounds of interest. Thus, PLE technology might revalue this low-quality fruiting body.

Depending on the pressurized solvent, molecules with a different lipophilic/hydrophilic nature could be extracted. Water was used to obtain high polysaccharides yields from several mushroom species and from *Tuber melanosporum* [7,24,25], whereas pressurized ethanol extracts certain lipids and fungal sterols [7,15]. The combination of the two solvents might stimulate a more selective extraction with both lipophilic and hydrophilic characteristics that might exert interesting or synergistic bioactivities. Therefore, in this work, a PLE method was designed to obtain bioactive fractions from two truffle species, summer truffle (*Tuber aestivum* Vittad.) and desert truffle (*Terfezia claveryi* Chatin), using a response surface methodology (RSM) as an optimization methodology. The bioactive compound levels present in the different fractions were quantified, and their ability to act against cell oxidation as well as an immunomodulator role were studied. Their potential against diabetes was also evaluated during metabolic syndrome-associated enzymes inhibition (α–amylase, α–glucosidase).

## 2. Materials and Methods

### 2.1. Biological Material

*Terfezia claveryi* truffles were harvested from the experimental station at the University of Murcia (Espinardo, Spain), and *Tuber aestivum* fruiting bodies were harvested in Gúdar-Javalambre forests (Teruel, Spain). Fresh truffles were identified, selected and processed according to Rivera et al. (2011) [26]. After, truffle samples (500 g) were lyophilized (LyoBeta 15 lyophilizer (Telstar, Madrid, Spain)), ground, mixed and sieved until a particle size lower than 0.5 mm was obtained. Powdered truffles were kept frozen (at −80 °C) until further use.

### 2.2. Reagents

Hexane (95%), HPLC-grade chloroform, methanol and acetronitrile solvents were obtained from LAB-SCAN (Gliwice, Poland). Ethanol (100%), sulfuric acid (H_2_SO_4_), and sodium carbonate (Na_2_CO_3_) were obtained from Panreac (Barcelona, Spain). Ascorbic acid, ergosterol (95%), D-glucose, gallic acid, and D-glucosamine hydrochloride standards were purchased from Sigma-Aldrich (Madrid, Spain). Potassium hydroxide (KOH), 2,6-Di-tert-butyl-*p*-cresol (BHT), hexadecane, bovine serum albumin (BSA), p-dimethylaminebenzaldehyde, HCl (37%), acetylacetone, phenol sulfuric were also from Sigma-Aldrich. All reagents and solvents used were of analytical grade.

### 2.3. Pressurized Liquid Extractions

Powdered *T. aestivum* and *T. claveryi* ascocarps (0.5 g) were submitted to PLE technology using an Accelerated Solvent Extractor (Dionex Corporation, ASE 350, USA). Before that, a mixture of truffle and washed sea sand (in a 1:8 ratio (*w:w*) (Panreac, Barcelona, Spain) was loaded in the extraction cell (10 mL) and covered with cellulose filters (Dionex Corporation, USA). Then, water (W), ethanol (100%) (E) and water:ethanol (1:1 *v/v*) (E:W) were selected as extraction solvents. Once the extraction cell was prepared, extraction procedures conditions were designed using RSM experimental methodology [25]. Fractions obtained using water were immediately frozen and afterward freeze-dried in a LyoBeta 15 lyophilizer (Telstar, Madrid, Spain) (final condenser temperature −80 °C) for 72 h. Those extracts obtained with ethanol submitted to a rotary vacuum evaporator at 40 °C (IKA^®^ RV 10, VWR International, Spain) in order to dry them. The E:W extracts were dried and then lyophilized. Afterward, all samples were stored at −20 °C.

A full factorial with three-level experimental design (3^2^) was used following RSM methodology in order to optimize PLE. Temperature (50, 115, and 180 °C) and time (5, 17.5, and 30 min) were selected as independent factors. The response variables investigated were the total carbohydrates (TCH), β-glucan, chitin, ergosterol, total phenolic compounds (TPC) as well as the total extraction yield obtained extraction yield. In total, eleven experiments were carried out in a randomized order, as indicated in Tejedor-Calvo et al. (2020) [7]. Nine points of the factorial design and two additional center points were selected to consider the experimental errors. All the experiments were carried out with three different solvents (W, E, E:W).

### 2.4. Determination of Truffles and PLE Extracts Composition

Total carbohydrate content of truffle fruiting bodies and extracts obtained using PLE (50 mG/mL) was quantified by the phenol–sulfuric acid method [27]. D-glucose curve standard was used for quantification. Chitin content (10 mG/mL) was quantified as described by Tejedor-Calvo et al. (2019) [28] using a glucosamine hydrochloride standard curve. Total β-glucan content (50 mg) was evaluated by a β-glucan determination kit specific for mushrooms and yeasts (Megazyme^®^, Biocom, Barcelona, Spain) following the instructions of the user’s manual. Soluble protein concentration (10 mG/mL) was determined using the Bradford method reagents (Sigma-Aldrich, Madrid, Spain) according to the instruction manual. BSA was used as standard for protein quantification. Total phenolic compound levels (10 mG/mL) were evaluated by the Folin–Ciocalteu method [29]. Gallic acid was used as standard for quantification. Truffle fruiting bodies and extracts obtained using PLE were saponified and analyzed by GC-MS-FID [28]. Ergosterol was used as standard, and Hexadecane (10% *v/v*) was used as internal standard and ergosterol as standard for ergosterol and derivative compounds quantification. The compound determinations were carried out in triplicate.

### 2.5. Testing of Cellular Antioxidant Activity (CAA)

Human colorectal adenocarcinoma cell lines Caco-2 (ATCC, Manassas, VA) were cultured in Dulbecco’s Modified Eagle’s Medium (DMEM) supplemented with 10% fetal bovine serum (FBS), 100 mG/mL streptomycin, 100 U/mL penicillin, 1% nonessential amino acids and 2 mM L-glutamine (Gibco, Paisley, U.K.), and incubated at 37 °C in humidified atmosphere containing 5% CO_2_.

First, the cytotoxicity of the truffle extract was evaluated in Caco-2 cells using the MTT test [30]. Afterward, cellular antioxidant activity was measured following the method described by Wolfe and Liu (2007) [31] with some modifications. Briefly, the Caco-2 cells (1.5 × 10^5^ cell/mL) were seeded in 96-well plates with complete medium. After 48 h, the medium was discarded, and the Caco-2 cells were washed with phosphate buffered solution (PBS). Cells were incubated at 37 °C for 1 h with the extracts in subtoxic concentrations and 25 µM of fluorescent marker DCFH-DA (2′,7′-Dichlorofluorescin diacetate) dissolved in fresh serum-free medium. The media were then removed, and cells were washed 3 × with PBS. Then, 600 µM of the free radical initiator ABAP (2,2′-Azobis(2-methylpropionamidine) dihydrochloride) in Hanks’ balanced salt solution (HBSS) were added to each well, and the plate was transferred to a plate reader Cytation 5 (Biotek) for measurements. Fluorescence readings were taken every 5 min for 1 h at excitation/emission wavelengths of 485/538 nm, for a total of 13 cycles. For quantification, a curve was plotted using the fluorescent measures using the area under the curve (AUC) calculated by the plate reader software. The reduction of fluorescence was calculated by triplicate for each concentration against the blank as follows:(1)$$\%\;inhibition=\left(1-\frac{AUCsample}{AUCblank}\right)\cdot 100$$

Once the percentage inhibition was established, the equation was determined and the 50% of inhibition was established as the IC_50_ value.

### 2.6. Testing of the Immunomodulatory Properties

The immunomodulatory properties were analyzed as described by Tejedor-Calvo et al. (2020) [7] with modifications. After 24 h of cell incubation, cells supernatants were selected and stored at −20 °C. A selection of pro-inflammatory cytokines TNFα (Tumor necrosis factor alpha), IL-1β (Interleukin 1 β) and IL-6 (Interleukin 6) were measured in the supernatants using a BD Biosciences Human ELISA set (Aalst, Belgium) following the manufacturer’s instructions. Then, the OD was measured at 450 nm using a multiscanner autoreader (Infinite M200, Tecan, Barcelona, Spain). The positive controls (cells stimulated with LPS) were considered as 100% cytokine secretion. The experiments were carried out in triplicate and results were presented as inhibition percentage.

### 2.7. Testing of α-Glucosidase and α-Amylase Inhibitory Activity

Truffes extracts (10 mg) obtained by PLE at the optimal extraction conditions using water, ethanol:water and ethanol as solvent were mixed with the same solvent, stirred in a Vortex for 2 min and centrifuged at 12,000 rpm. Obtained supernatants were used as a source of potential inhibitors.

The extract ability to inhibit the key enzymes involved in carbohydrate digestion was determined by adapting the concentrations described in the methods for α–glucosidase [32] and α–amylase [33] to fit in the absorbance range and to obtain an effective reaction timing. Briefly, truffle supernatants (10 µL) or arcabose (1 mG/mL) were mixed with 20 µL α–glucosidase in 100 mM sodium phosphate buffer pH 6.9 and incubated for 5 min. Afterward, PNPG (2 mG/mL) was added (200 µL), and the reaction was spectrofotometrically followed (Genesys 10-S, Thermo Fisher scientific) at 400 nm and 37 °C during 10 min. Similarly, truffle supernatants (20 µL) were mixed with 100 µL α–amylase (1 mG/mL 100 mM sodium phosphate buffer pH 6.9) and 100 µL starch 1% and incubated at 20 °C during 3 min. Then, DNS (100 µL) was added, and the mixture was heated at 100 °C for 15 min and diluted with 900 µL MilliQ water. The enzymatic reaction was followed for 10 min while preparing the tubes with 2 min interval. The absorbance changes were measured at 540 nm. All assays were performed in duplicate. Arcabose and the extracts were tested in different concentrations to establish their IC_50_ value.

### 2.8. Statistical Analysis

Differences values were evaluated at a 95% confidence level (*p* ≤  0.05) using a one-way analysis of variance (ANOVA) followed by Tukey’s multiple comparison test.

Optimal PLE conditions were selected using multiple linear regressions with Statgraphics Centurion XVI software (Statpoint Technologies, Warrenton, VA, USA). The statistical analyses and graphs were performed using GraphPadPrism version 5.01 for Windows (GraphPad Software, San Diego, CA, USA). Principal component analysis (PCA) was also performed and visualized in RStudio 1 February, 1335 (Rstudio Team, 2019) using R version 3.6.1. For each model, the ANOVA assumptions were assessed through the Levene test (homogeneity of variance) and the Shapiro–Wilk test (normality).

## 3. Results and Discussion

Two truffles of *Tuberaceae* and *Pezizaceae* families were selected to optimize PLE extraction methods because larger differences were expected between them, as in previous works [6]. Nevertheless, a preliminary analysis of their major constituents and other particular bioactive compounds was carried out to compare them with other truffles strains described in previous reports [7].

### 3.1. Chemical Composition of Truffles

Both truffle species showed large TCH contents, with those of *T. claveryi* slightly higher than *T. aestivum* (Table 1), but lower than those reported for *Terfezia* (46–48%) and *Tirmania* (53–83%) [34,35,36]. *T. aestivum* TCH content however was in the range of levels observed for other *Tuber* truffles (31–36%) [7]. In both species, the main carbohydrates were β-glucans, followed by chitin, since together they were approx. 89–95% of their TCH levels. Thus, *T. aestivum* reported lower a β-glucan concentration, but a slightly higher chitin content than *T. claveryi*. These values were similar to those previously reported for *T. aestivum* (13.3% chitin and 24.5% β-glucans) [37]. Summer truffle protein content was also in concordance with previous studies (9–11%) [37,38], while they were lower in comparison with those reports in *T. claveryi* (32–35%) [36,39]. The slight differences might be due to the nutritional status of the ascocarp, to environmental conditions or to the developmental stage as stated by [40]. In addition, they might also differ because of variations in the analytical method used. Phenolic compound levels ranged according to previous studies (0.6 to 1.1 mg/g) [7], and they might be more variable because they are synthetized during secondary metabolism.

The ascocarps of *T. claveryi* and *T. aestivum* showed higher sterol total levels than phenolic compound amounts. (Table 1). Ergosterol and brassicasterol were the two main sterols, followed by ergosta7,22-dienol. The brassicasterol content (29% of the total sterols) was in line with previous studies of the *Tuber* genus, in which brassicasterol ranges were between 28–44% depending on the species. In *Terfezia* truffles, brassicasterol levels were 98% of total sterols, whereas ergosterol was present in lower amounts [11]. However, ergosterol as the main constituent of fungal hyphae, was used as a biomarker [41], and therefore, high ergosterol levels indicate proper fungal growth. The stigmasterol compound was only detected in *T. aestivum*, as noted in previous studies [7]. Sommer and Vetter (2020) [42] reported 25 different sterols in *T. aestivum* and *Tuber borchii*, but they were present in low concentrations. The detection of certain sterols typical from plants (e.g., stigmasterol or brassicasterol) might be due to the nutrient exchange between truffles and the host plants with which they associate [11].

Therefore, since the major difference between both truffles was the structural polysaccharide content, the PLE methods were separately optimized for both species because a different sample texture might influence the extraction yields.

### 3.2. Optimization of PLE Extraction Methods

Truffles were submitted to PLE following a full factorial 3^2^ experimental design. The extraction was carried out using pressurized W, E and E:W (1:1) solvents, the conditions covering nearly the full operational range but within the temperature and pressure limits of the equipment. The design was adjusted to maximize the amount of bioactive compounds extracted by considering the variable yield and metabolite content, both equally important, and temperature and extraction time as independent factors (Table 2 and Table 3). An ANOVA test was performed for each response to establish the optimal statistical model that fits to the desirability function, allowing for a simultaneous optimization of several responses.

#### 3.2.1. Carbohydrate-Enriched Fractions

Temperature, more than time, positively influenced the extraction yields in both truffles independent of the solvent tested (Figure 1 and Appendix A. A similar pattern was previously noted in *T. melanosporum* and mushroom β-glucans extraction [7,25]. The yields obtained using water as a solvent (Table 2 and Table 3) were 7.3% and 5.3% higher than those noted in black truffle with the same extraction conditions (180 °C–30 min) [7]. However, lower yields were obtained when both truffles were submitted to E:W (1:1), and particularly to ethanol. Yields for *T. aestivum* were similar to those obtained for *T. melanosporum* (22.5%) when ethanol was applied [7] but were lower than *T. claveryi*. In addition, using an E:W (1:1) and low temperature (50 °C), only 9–11% material was extracted from *T. claveryi* compared to 29–31% from *T. aestivum*. Apparently, the extraction of polysaccharides and other more apolar compounds required stronger conditions to release them from the fungal matrix. Similar results were noted when *T. melanosporum* was submitted to PLE extractions [7]. Furthermore, extraction yield differences between truffles might be due to β-glucans levels present in their hyphal walls. *T. aestivum* contained less β-glucans levels, whereas *T. claveryi* reported higher chitin concentrations. These results suggested that β-glucans (more than chitin) might impair extractions not only of polysaccharides but also of other constituents.

The response surface analysis pointed out 115 °C for 17.5 min using water as the optimal conditions to extract polysaccharides (TCH and β-glucans) enriched fractions from *T. claveryi* (Figure 1). However, stronger conditions (180 °C for 30 min) were needed to efficiently extract fractions with a high chitin concentration (Table 2). Similar results were obtained for *T. aestivum* (optimal conditions: 180 °C for 30 min) (Table 3), and in concordance with a previous study for *T. melanosporum* [7]. Under these conditions, the largest yields of TCH were obtained, 71.3% and 69.3% for *T. claveryi* and *T. aestivum,* respectively, as well as the highest concentrations of β-glucan (27.18% and 15.48%) and chitin (7.66% and 8.70%). When the intermediate temperature (115 °C) was applied, the TCH content was similar compared to those obtained with higher temperatures (180 °C), while β-glucan and chitin content were lower. Perhaps moderate temperatures might extract other compounds more efficiently such as mono-or disaccharides that might be destroyed with higher temperatures. RSM plots showed that β-glucans were easier to extract from *T. claveryi* than from *T. aestivum* (Figure 1A). Similar behavior was noted for chitin (Figure 1B). Extraction of structural polysaccharides from cell walls might be dependent on several factors such as their concentration, the type and number of bounds between them, and their solubility in the solvent used for extraction, among others [24,25]. According to these results, long extractions (30 min) with water (180 °C) were more adequate to obtain enriched polysaccharide fractions (including β-glucan and chitin in higher levels) from both truffle species. The addition of ethanol to the extraction solvent significantly reduced the amount of all studied carbohydrates, probably because this solvent reduces their solubility in water (induces their precipitation), as it is used to isolate them from complex mixtures [43].

#### 3.2.2. Protein-Enriched Fractions

PLE did not efficiently extract proteins from truffles under the tested conditions (less than 1% of the soluble protein levels noted in the ascocarps) (Table 1). Short and mild extraction conditions (50 °C, 5 min) using water could extract more proteins from *T. claveryi* compared to slightly higher temperatures and longer extraction times. These results suggest that the selected conditions were too strong to extract proteins. These conditions probably broke down many of the free proteins or bound them to polysaccharides (i.e., by Maillard reactions) since the protein levels slightly increased again with the highest tested temperatures [44]. At these temperatures, some bounds might also break, releasing still too small protein levels. Therefore, these PLE conditions could be useful to precipitate or eliminate proteins from carbohydrate-enriched fractions by obtaining extracts where polysaccharides are the major constituents. The extraction using E:W mixture did not improve the protein extraction compared with 100% water. Ethanol induced the denaturation of fungal proteins.

#### 3.2.3. Phenolic-Enriched Fractions

The optimal conditions to obtain TPC-enriched fractions were 180 °C for 30 min (Figure 1C). Water was selected as the best solvent to extract them, despite that high amounts using ethanol and E:W were also obtained. Apparently, the use of PLE helped to release bound phenolic compounds that went undetected when they were directly determined from the powdered truffles. *T. claveryi* aqueous extract, for instance, contained 3.06 mg/g of extract, and its extraction yield was 71.3% indicating that approx. 2.18 mg phenolic compounds were extracted per gram of truffle. According to the results obtained (Table 1), *T. claveryi* contained only 1.02 mg/g, and therefore, PLE can be pointed to as an interesting tool to obtain highly concentrated fractions of phenolic compounds.

#### 3.2.4. Sterol-Enriched Fractions

The same optimal conditions parameters (30 min—180 °C), except for the solvent, were necessary to obtain total sterol enrichment. Ethanol solvent was more suitable compared with water or E:W mixture. Among sterols, traces of 9,19-cyclolanost-7-en-3-ol were only detected in *T. aestivum*. The response surface plots revealed significantly lower desirability values for *T. claveryi* extraction than for *T. aestivum* (Figure 1E). In general, higher temperatures and times slightly increased sterols levels, except for ergosterol in *T. claveryi* (115 °C and 17.5 min showed apparently adequate). These results were in line with those from Gil-Ramírez et al. (2013) [45], who found that at 100 °C for 5 min (1 cycle) extraction conditions yielded higher sterol levels from *A. bisporus* than longer extraction times and cycles, and those of Tejedor-Calvo et al. (2020) [7] (optimal conditions: 115 °C for 17.5 to 30 min). Similarly, concentrations of ergosta7.22-dienol were higher at 115 °C compared with higher temperatures, suggesting that PLE might induce partial transformation of ergosterol into this and other derivatives. In both truffle species, brassicasterol and stigmasterol levels in the extracts were slightly enhanced when temperature and time were increased. High brassicasterol levels were found versus ergosterol in some extracts, promoting the possibility of ergosterol transformation into other derivatives due to intense PLE conditions.

#### 3.2.5. Multivariate Data Analysis of Extracted Compounds

Principal components analysis (PCA) was used to explore the possible correlations of PLE conditions with truffle extracted compounds (Figure 2). The PCA analysis explained 88.5% of the data variability with the two first components. The first component alone explained 78.8% of the variability, which was mainly linked to variability in yield and extracted quantities of TCH, chitin, proteins, TPC and β-glucans (all of them showed negative loadings: −0.40, −0.41, −0.40, −0.39, −0.39 and −0.38) (Figure 2A). The second component explained an additional 9.7% of data variability and was mainly associated with the extracted sterols quantity, which was the only variable with a high absolute value of loading (−0.91) (Figure 2A). The PCA allowed for the clear separation of all the samples into four clusters (Figure 2B): the west group (yellow) grouped water extracts, with either high temperature (HT) or intermediate temperature (IT) (including both species for HT and only *T. claveryi* for IT). This cluster was clearly associated with high yields and high extracted quantities of TCH, chitin, proteins, TPC and β-glucans (Figure 2A). On the contrary, the down-east group (blue) was associated with high extracted quantities of sterols, including all ethanol samples except for the low temperature (LT) *T. aestivum* samples (Figure 2B). The remaining clusters (orange and green) lay in intermediate positions.

In order to confirm these linkages, a permutational multivariate analysis of variance (permANOVA) was applied on the Euclidean distance among samples. The extraction solvent was the predictor that best correlated with data variability (*p* < 0.001, explained R^2^ = 0.60), with water extraction correlating to high yield values and extracted quantities of TCH, chitin, proteins, TPC and β-glucans; and ethanol extraction correlating to high sterol values. The extraction temperature also explained a significant part of data variability (*p* < 0.001, explained R^2^ = 0.14), with temperature positively correlating to high yield values and extracted quantities of TCH, chitin, proteins, TPC and β-glucans. The permANOVA showed significant differences between truffles, although it only explained a low share of data variability (*p* < 0.001, explained R^2^ = 0.05). *T. claveryi* was correlated with slightly higher yield values and extracted quantities of TCH, chitin, proteins, TPC and β-glucans when compared with *T. aestivum*, probably because of differences in cell wall composition. No significant effects of extraction time were found (*p* = 0.46).

According to the set of results obtained from the response surface plots together with PCA analysis, the combination of 180 °C and 30 min was selected as the optimal extraction conditions (higher yields). Although these conditions were more effective for *Terfezia* truffles than for the *Tuber* genus [7]. Water was pointed to as the more adequate solvent to extract major polysaccharides and phenolic compounds, whereas ethanol was selected to obtain total sterol-enriched fractions. With a view to detect possible synergies, the extracts obtained under those conditions but using the three solvents were further tested to investigate their potential biological properties.

### 3.3. Antioxidant Properties

The antioxidant activities of some truffles species such as *Tuber indicum, Tuber magnatum, and Picoa juniperi,* among others, were evaluated in previous studies but usually using in vitro colorimetric tests (DPPH, ABTS, etc.) [34,46,47,48]. These assays might be interesting as preliminary screening through many samples, but their biological significance is limited since antioxidants must pass through the cell membrane and reach their targets. Therefore, the antioxidant potential of *T. claveryi* and *T. aestivum* extracts was studied using cell cultures. Results indicated that they contained metabolites able to enter the cell and protected it from free radicals such as those initiated by ABAP (Table 4). Within the *T. claveryi* PLE extracts, those extracted using W were more effective than E:W extracts. The latter was more effective than the E extract, thus suggesting that the main antioxidant agents might be water soluble polar compounds. The antioxidant mechanism of *T. aestivum* compounds might be differ from the desert truffle working. Although its W and E extracts were less effective than those from *T. claveryi*, the use of an E:W mixture generated a fraction with higher antioxidant activity than the other solvents. This fact suggests that synergistic mechanisms might be involved within the polar and less polar antioxidants. Nevertheless, the EC_50_ values obtained were higher than those obtained with E extracts from mushrooms (182 mG/mL) [49], while they were lower than some plant extracts obtained from cauliflower, tomato, lettuce, etc.

Previous studies carried out on *T. claveryi* and *P. juniperi* pointed to the raw material (without industrial processing) as those responsible for the main antioxidant activity of these truffles. They reported the ability to inhibit lipid peroxidation (LOO^●^), deoxyribose (OH^●^), and peroxidase (H_2_O_2_) [50]. However, other publications suggested that phenolic compounds were those responsible for antioxidant activities, such as in methanolic extracts obtained from *T. magnatum* [6]. The PLE water extracts from *T. claveryi* and *T. aestivum* contained higher TPC concentrations than those obtained with ethanol as solvent, showing lower EC_50_. However, the E:W extract (lower TPC levels) presented an EC_50_ of only slightly higher than the water extract in the case of *T. claveryi* and lower in *T. aestivum* (Table 4). These observations might indicate two possibilities: (i) that in these truffles, not all the phenolic compounds are antioxidants, with the activity being due to a particular compound present in low concentrations but that is powerful (and probably better solubilized in E:W mixture); or (ii) that despite the lower TPC concentration found in the E:W extracts, synergetic reactions are improving the antioxidant capacity of these extracts (as noted particularly for *T. aestivum* extracts).

### 3.4. Immunomodulatory Properties

The obtained PLE extracts were also applied to macrophages differentiated from THP-1 human monocytes cultures to study their ability to modulate macrophage response to LPS, since previous reports indicated that the β-glucans from a *T. melanosporum* extract were able to reduce secretions of pro-inflammatory cytokines [7].

The cytotoxicity experiments indicated that when the PLE extracts were applied up to 15 μg/mL (*T. aestivum*) and 40 μg/mL (*T. claveryi*), the THP-1 macrophages’ viability was not affected (data not shown). Therefore, W, E:W and E extracts were applied at those concentrations.

The THP-1 macrophages stimulated with LPS (positive control) showed a significant release of IL-6, IL-1β and TNFα (pro-inflammatory cytokines) in comparison with non-stimulated cells (negative control) (Figure 3). The addition of the PLE extracts influenced the amount of interleukine-6 liberated in the media independently of the solvent or truffle species used, showing a release of approx. 50% less than the positive control. However, the extracts obtained from *T. claveryi* were not able to inhibit IL-1β, and those obtained from *T. aestivum* showed lower inhibition levels than noted for IL-6. The E extracts from both truffles showed completely different behavior. *T. aestivum* extract did not influence TNFα release, whereas the one from *T. claveryi* reduced the secretion found in the positive control to approx. 80%. The other extracts obtained with W or E:W from both truffles also modulated the secretion of TNFα, but in low amounts compared to E extract response from *T. claveryi*. These results suggest that the anti-inflammatory modulation induced on the macrophage response to LPS by the two truffles extracts might occur via different mechanisms, perhaps because different compounds might be involved. PLE extracts obtained from *T. melanosporum* reduced approx. 20% and 40% the release of IL-6 and IL-1β, respectively, showing no effect against TNFα secretion [7]. The effect noted with W or E:W extracts was not tested for *T. melanosporum*. Therefore, similar behavior could be suggested for both truffles from the same genus. The β-glucans present in the *T. melanosporum* PLE extract were pointed to as the responsible agent for the modulation of the cytokine secretion. However, the *T. aestivum* PLE extract obtained with ethanol was also effective, suggesting that other compounds might also be involved in the anti-inflammatory activity noted. Moreover, the concentrations of the PLE extracts applied to the macrophages (particularly those obtained from *T. aestivum*) were almost 10 times lower than other PLE fractions extracted from other edible mushroom species (150 µg/mL) [51]. This suggested that they might have more of an effect than other fungal extracts. Nevertheless, further in vivo studies are needed to confirm these observations.

### 3.5. Amylase and Glucosidase Inhibitory Activities

The inhibition of the α–amylase and α–glucosidase activities facilitates the maintenance of circulating glucose levels by decreasing the rate of blood sugar absorption, and therefore, inhibitors such as acarbose are used to prevent diabetes or other metabolic disorders [52]. PLE extracts obtained using water as solvent from *T. claveryi* showed 10-fold higher amylase inhibitory capacity to acarbose (Table 5). However, *T. aestivum* W extracts showed a closer inhibitory activity (IC_50_: 9.44 mG/mL). The *T. aestivum* E extract was also more effective compared to *T. claveryi*. The extracts obtained with a combination of both solvents did not significantly improve their capacity to inhibit α–amylase. Within the extracts obtained from *T. claveryi*, glucosidase inhibition was only observed in E:W and E extracts, showing IC_50_ values similar and even lower with respect to the control. The *T. aestivum* E extract was also more effective than its water extract but less than the one from *T. claveryi*. A recent study reported lower IC_50_ values for *T. aestivum* methanolic extracts than the PLE extracts using ethanol (0.49 mG/mL) for α–amylase, while they were similar for α–glucosidase (8.54 mG/mL) [53]. The parameters used (temperature, solvent or pressure) for PLE extraction might be detrimental to obtain α–amylase inhibitors or the differences within the strain or developmental stage might influence their presence. The IC_50_ values obtained ranged according to some medicinal and edible mushrooms extracts, with IC_50_ values between 0.71–1.72 mG/mL (i.e., *Agaricus blazei*, *Coprinus comatus*, *Morchella conica*) [54]. According to a previous reports, polysaccharides [55,56], fatty acids [57,58], as well as phenolic compounds or polyphenols [59] might be the responsible compounds for the noted hypoglycemic activities. Therefore, the fungal polysaccharides present in the W extracts and the phenolic compounds of the E extracts might be related to the inhibition noted. However, further in vivo experiments are necessary to confirm in vitro assays.

## 4. Conclusions

*T. aestivum* and *T. claveryi* truffles are potential sources of bioactive compounds. PLE methodology was able to obtain β-glucan-, chitin-and phenolic-enriched fractions using water and fungal-enriched fractions with ethanol. Temperature is a key factor, compared with time, on the compounds collected. Optimal extraction conditions (30 min, 180 °C) generated fractions with potential interests on antioxidant, immunomodulatory and hypoglycemic activities. The vitro tests suggested that it might be worth carrying out further in vivo experiments. Unfortunately, the use of a mixture of ethanol:water as an extraction solvent did not induce synergism in the determined biological properties.

## Figures and Tables

**Figure 1 foods-11-00298-f001:**
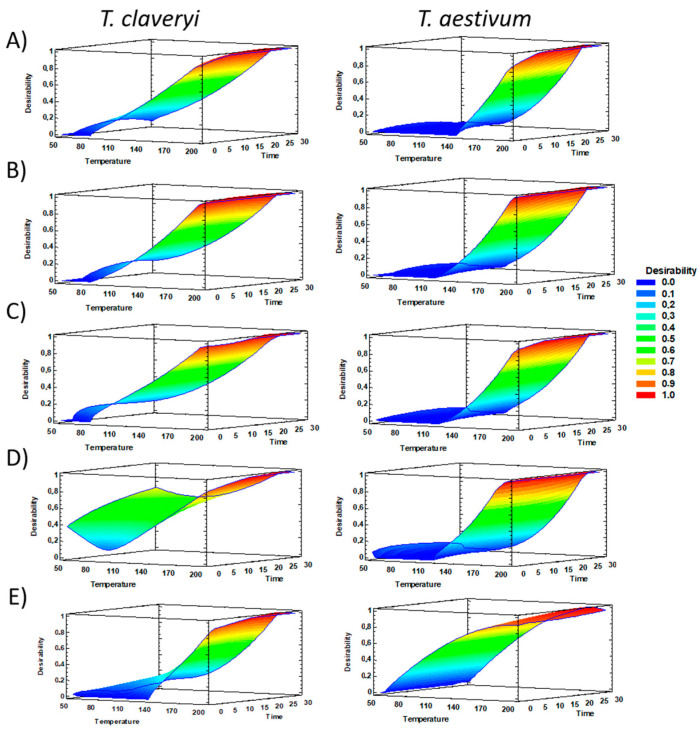
Response surface 3D plots (desirability function) of PLE extractions from *T. claveryi* (Tc) and *T. aestivum* (Ta). The response variables investigated were the content in the obtained fractions of (**A**) β-glucans, (**B**) chitin, (**C**) proteins, and (**D**) TPC extracted using water as solvent and (**E**) total sterols extracted using ethanol as solvent.

**Figure 2 foods-11-00298-f002:**
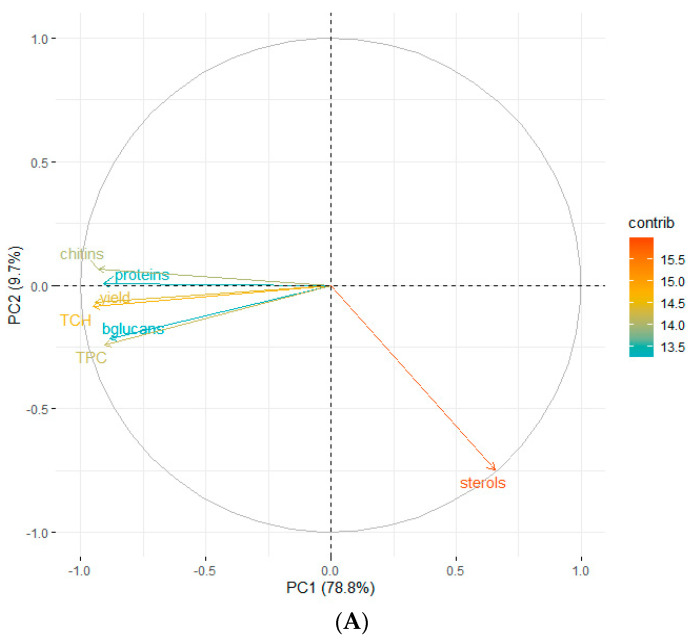
PCA (**A**) loading plot for compounds extracted by PLE and (**B**) score plot for compound variation among PLE samples. Samples names (extraction conditions) are those indicated in Table 1 and Table 2. Arrow color indicates the contribution of a compound to the PCA components (contrib) and sample color indicates the quality of representation for the sample (cos2). Ter: *Terfezia claveryi*; Tub: *Tuber aestivum*; HT: high temperature (180 °C); IT: intermediate temperature (115 °C); LT: low temperature (50 °C); lt: long time (30 min), it: intermediate time (17.5 min), st: short time (5 min); W: water; E: ethanol; WE: water–ethanol mixture (1:1).

**Figure 3 foods-11-00298-f003:**
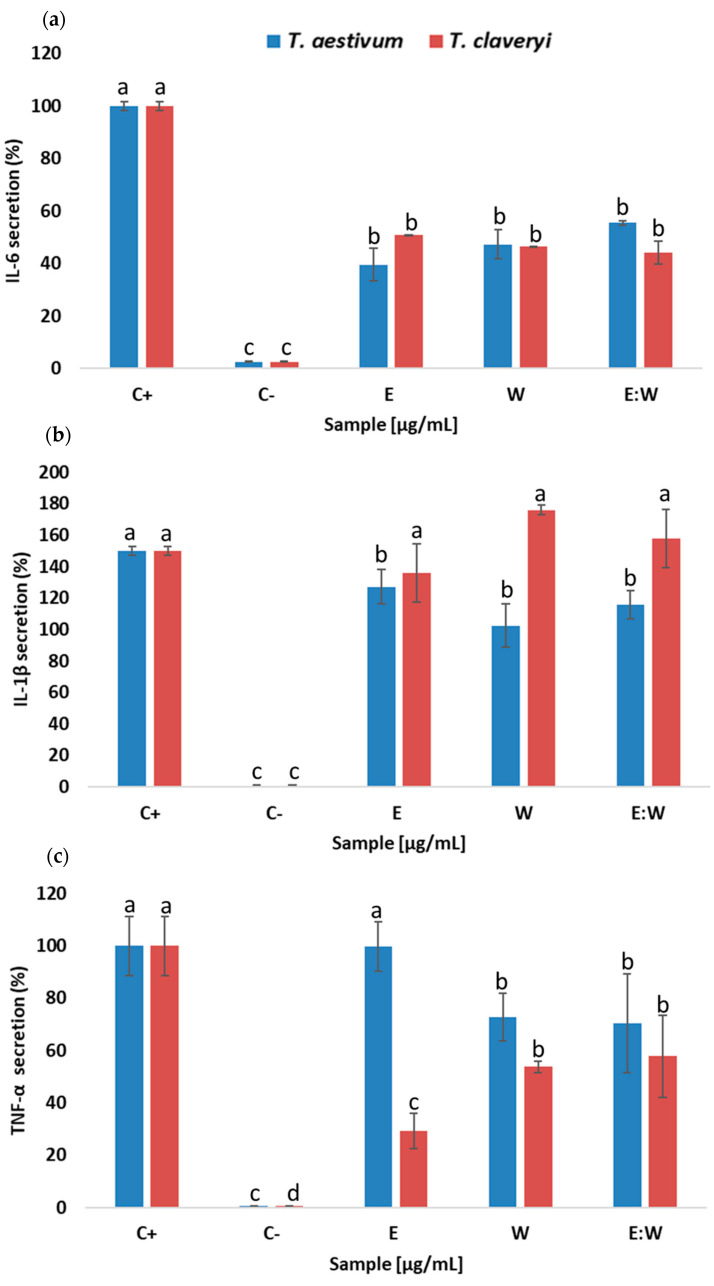
Levels of (**a**) IL-6, (**b**) IL-1β and (**c**) TNFα secreted by THP-1/M activated with LPS in the presence of PLE extracts obtained from truffles using ethanol (E), water (W), and water:ethanol mixture (1:1) (E:W) after 30 min extraction at 180 °C. Positive control (cells stimulated with LPS, C+); negative control (cells without LPS, C-). Each bar is the mean of three determinations ± SD. ^a,b,c,d^ Different letters denote significant differences (*p* ≤ 0.05) between different truffle extracts (*p* ≤ 0.05). Equations are shown in Appendix A.

**Table 1 foods-11-00298-t001:** Total carbohydrates, total proteins, total phenolic compounds (TPC), and sterols in *T. claveryi* and *T. aestivum* truffles. Values are the mean of replicates, and error bars indicate standard deviations. Indicated values are *w/w* ^a,b^.

	*Terfezia claveryi*	*Tuber aestivum*
Carbohydrates (g/100 g)	38.44 ± 1.35 ^a^	35.83 ± 0.70 ^b^
β-Glucans (g/100 g)	27.96 ± 1.55 ^a^	21.71 ± 0.43 ^b^
Chitin (g/100 g)	8.53 ± 0.26 ^a^	10.20 ± 0.63 ^a^
Proteins (g/100 g)	8.92 ± 1.05 ^a^	11.92 ± 0.60 ^a^
TPC (mg/g)	1.02 ± 0.07 ^a^	1.04 ± 0.04 ^a^
Ergosterol (mg/g)	2.30 ± 0.23 ^a^	2.27 ± 0.40 ^a^
Brassicasterol (mg/g)	1.40 ± 0.15 ^b^	1.71 ± 0.13 ^a^
Ergosta7.22-dienol (mg/g)	1.16 ± 0.02 ^a^	1.09 ± 0.03 ^a^
Stigmasterol (mg/g)	n.d.	0.65 ± 0.04

n.d., not detected. Different letters denote significant differences (*p* ≤ 0.05) between different truffle species (*p* ≤ 0.05).

**Table 2 foods-11-00298-t002:** Yields and concentrations of the main compounds extracted by PLE following a full factorial 3^2^ experimental design from *Terfezia claveryi*
^a,b,c,d,e^.

Independent Factors			Investigated Responses																							
			Yield (% Truffle)			TCH (g/100 g Extract)		β-Glucans (g/100 g Extract)		Chitin (g/100 g Extract)		Soluble Rroteins (g/100 g Extract)		TPC (mg/g Extract)			Total Sterols (mg/g Extract)		Ergosterol (mg/g Extract)		Brassicasterol (mg/g Extract)		Ergosta7.22-dienol (mg/g Extract)		Stigmasterol (mg/g Extract)	
Run	Temperature (^°^C)	Time	W	E	E:W	W	E:W	W	E:W	W	E:W	W	E:W	W	E	E:W	E	E:W	E	E:W	E	E:W	E	E:W	E	E:W
1	50	5	39.4	7.6	9.3	43.15 ± 4.43 ^b^	40.00 ± 2.24 ^a^	27.92 ± 1.11 ^b^	19.79 ± 0.35 ^a^	5.97 ± 0.20 ^d^	3.64 ± 0.10 ^d^	1.87 ± 0.01 ^b^	1.38 ± 0.15 ^c^	1.84 ± 0.18 ^b^	0.05 ± 0.04 ^d^	0.24 ±0.12 ^d^	2.56 ± 0.18 ^d^	n.d.	0.47 ± 0.10 ^e^	n.d.	2.09 ± 0.08 ^b^	n.d.	n.d.	n.d.	n.d.	n.d.
2	50	17.5	44.3	7.5	9.6	44.44 ± 3.78 ^b^	44.80 ± 3.54 ^a^	25.20 2.53± ^b^	23.34 ± 2.58 ^a^	6.53 ± 0.09 ^cd^	4.80 ± 0.10 ^c^	1.88 ± 0.22 ^b^	1.57 ± 0.03 ^c^	2.06 ± 0.32 ^ab^	0.08 ± 0.02 ^d^	0.29 ± 0.06 ^d^	2.50 ± 0.42 ^d^	n.d.	0.50 ± 0.08 ^e^	n.d.	2.00 ± 0.34 ^b^	n.d.	n.d.	n.d.	n.d.	n.d.
3	50	30	43.5	7.9	11.2	44.87 ± 2.61 ^b^	46.81 ± 3.03 ^a^	23.07 ± 1.57 ^b^	20.82 ± 0.23 ^a^	6.74 ± 0.06 ^d^	5.20 ± 0.89 ^c^	2.00 ± 0.03 ^ab^	1.34 ± 0.10 ^c^	2.32 ± 0.44 ^ab^	0.07 ± 0.14 ^d^	0.34 ± 0.04 ^d^	2.71 ± 0.39 ^d^	n.d.	0.93 ± 0.06 ^d^	n.d.	1.78 ± 0.33 ^b^	n.d.	n.d.	n.d.	n.d.	n.d.
4	115	5	49.4	13.9	35.9	50.00 ± 7.53 ^b^	22.67 ± 5.83 ^c^	32.28 ± 3.17 ^a^	16.03 ± 0.51 ^b^	7.05 ± 0.14 ^c^	7.91 ± 0.95 ^a^	1.84 ± 0.06 ^b^	1.11 ± 0.21 ^d^	1.43 ± 0.19 ^c^	0.31 ± 0.28 ^c^	0.81 ± 0.03 ^c^	3.74 ± 0.49 ^c^	0.13 ± 0.03 ^c^	1.41 ± 0.21 ^c^	n.d.	2.33 ± 0.25 ^b^	0.13 ± 0.03 ^b^	n.d.	n.d.	n.d.	n.d.
5	115	17.5	49.9	14.1	37.3	54.30 ± 2.84 ^ab^	31.84 ± 2.10 ^b^	35.54 ± 1.16 ª	14.58 ± 0.70 ^bc^	8.91 ± 0.23 ^b^	7.14 ± 0.37 ^ab^	2.18 ± 0.05 ^a^	2.13 ± 0.18 ^b^	2.26 ± 0.16 ª	0.40 ± 0.02 ^c^	0.88 ± 0.05 ^c^	4.27 ± 0.76 ^bc^	0.16 ± 0.04 ^c^	1.58 ± 0.32 ^c^	n.d.	2.20 ± 0.40 ^b^	0.16 ± 0.04 ^b^	0.48 ± 0.08 ^b^	n.d.	n.d.	n.d.
6	115	17.5	55.5	15.9	35.1	59.01 ± 2.18 ª	36.72 ± 1.88 ^b^	35.85 ± 1.86 ª	15.23 ± 0.93 ^b^	9.51 ± 0.56 ^b^	8.03 ± 0.57 ^a^	1.90 ± 0.17 ^b^	1.45 ± 0.21 ^c^	2.03 ± 0.07 ^a^	0.39 ± 0.02 ^c^	0.96 ± 0.01 ^c^	4.85 ± 0.53 ^bc^	0.12 ± 0.04 ^c^	1.96 ± 0.17 ^bc^	n.d.	2.45 ± 0.32 ^b^	0.12 ± 0.04 ^b^	0.44 ± 0.10 ^b^	n.d.	n.d.	n.d.
7	115	17.5	49.9	14.4	36.3	62.62 ± 3.86 ^a^	33.04 ± 3.01 ^b^	38.80 ± 2.54 ^a^	15.12 ± 0.60 ^b^	9.74 ± 0.52 ^b^	5.93 ± 0.41 ^bc^	2.34 ± 0.39 ^a^	1.52 ± 0.13 ^c^	1.88 ± 0.22 ^bc^	0.28 ± 0.06 ^c^	0.85 ± 0.23 ^c^	5.33 ± 0.45 ^b^	0.40 ± 0.08 ^b^	2.40 ± 0.23 ^b^	n.d.	2.46 ± 0.15 ^b^	0.28 ± 0.05 ^a^	0.47 ± 0.07 ^b^	0.12 ± 0.03 ^b^	n.d.	n.d.
8	115	30	49.8	17.4	35.7	53.23 ± 2.85 ^b^	36.91 ± 2.93 ^b^	38.74 ± 2.59 ^a^	16.99 ±0.22 ^b^	9.57 ± 0.80 ^b^	7.65 ± 0.24 ^a^	2.56 ± 0.16 ^a^	2.33 ± 0.27 ^b^	2.02 ± 0.51 ^bc^	0.37 ± 0.07 ^c^	0.88 ± 0.05 ^c^	5.82 ± 0.63 ^b^	0.50 ± 0.10 ^b^	2.64 ± 0.25 ^b^	n.d.	2.58 ± 0.26 ^b^	0.33 ± 0.06 ^a^	0.59 ± 0.12 ^b^	0.17 ± 0.04 ^ab^	n.d.	n.d.
9	180	5	68.5	24	45.1	50.92 ± 4.01 ^b^	36.40 ± 1.15 ^b^	35.58 ± 2.18 ^a^	13.38 ± 0.23 ^c^	10.69 ± 0.53 ^a^	6.32 ± 0.21 ^b^	2.04 ± 0.25 ^ab^	3.29 ± 0.34 ^a^	2.56 ± 0.57 ^a^	1.12 ± 0.20 ^b^	1.34 ± 0.34 ^b^	11.36 ± 0.97 ^a^	0.46 ± 0.04 ^b^	4.70 ± 0.53 ª	n.d.	5.38 ± 0.34 ^a^	0.32 ± 0.02 ^a^	0.47 ± 0.10 ^b^	0.13 ± 0.02 ^b^	0.81 ± 0.10 ^b^	n.d.
10	180	17.5	72.5	28.9	45.4	49.04 ± 4.74 ^b^	37.08 ± 1.14 ^b^	36.12 ± 2.75 ^a^	14.23 ± 0.60 ^bc^	10.06 ± 0.25 ^a^	6.75 ± 0.26 ^b^	1.87 ± 0.26 ^b^	3.64 ± 0.18 ^a^	2.38 ± 0.32 ^ab^	1.16 ± 0.68 ^ab^	1.60 ± 0.48 ^ab^	12.49 ± 0.79 ^a^	0.71 ± 0.16 ^a^	4.93 ± 0.34 ^a^	0.10 ±0.03 ^a^	5.54 ± 0.27 ^a^	0.42 ± 0.08 ^a^	1.00 ± 0.18 ^a^	0.20 ± 0.05 ^ab^	1.02 ± 0.15 ^ab^	n.d.
11	180	30	71.3	32.4	45.7	48.86 ± 2.72 ^b^	37.24 ± 1.30 ^b^	38.12 ± 2.30 ^a^	13.35 ± 0.72 ^c^	10.74 ± 0.54 ^a^	7.82 ± 0.58 ^a^	2.03 ± 0.14 ^ab^	2.36 ± 0.26 ^b^	2.66 ± 0.05 ^a^	1.66 ± 0.05 ^a^	1.92 ± 0.04 ^a^	12.84 ± 0.98 ^a^	0.76 ± 0.14 ^a^	5.64 ± 0.48 ^a^	0.15 ± 0.04 ^a^	5.04 ± 0.40 ^a^	0.40 ± 0.07 ^a^	1.05 ± 0.10 ª	0.21 ± 0.03 ^a^	1.11 ± 0.08 ^a^	n.d.

n.d. not detected. Different letters denote significant differences (*p* ≤ 0.05) between different truffle species (*p* ≤ 0.05).

**Table 3 foods-11-00298-t003:** Yields and concentrations of the main compounds extracted by PLE following a full factorial 3^2^ experimental design from *Tuber aestivum*
^a,b,c,d,e^.

Independent Factors			Investigated Responses																									
			Yield (% Truffle)			TCH (g/100 g Extract)		β-Glucans (g/100 g Extract)		Chitin (g/100 g Extract)		Soluble Proteins (g/100 g Extract)		TPC (mg/g Extract)			Total Sterols (mg/g Extract)		Ergosterol (mg/g Extract)		Brassicasterol (mg/g Extract)		Ergosta7.22-Dienol (mg/g Extract)		Stigmasterol (mg/g Extract)		9.19ciclolanost-7-en-3-ol (mg/g Extract)	
Run	Temperature (^°^C)	Time	W	E	E:W	W	E:W	W	E:W	W	E:W	W	E:W	W	E	E:W	E	E:W	E	E:W	E	E:W	E	E:W	E	E:W	E	E:W
1	50	5	42.03	2.85	30.87	20.95 ± 3.21 ^e^	3.87 ± 0.26 ^e^	3.18 ± 0.45 ^d^	1.79 ± 0.26 ^e^	5.49 ± 0.64 ^e^	1.45 ± 0.23 ^d^	0.94 ± 0.12 ^a^	0.90 ± 0.21 ^e^	1.75 ± 0.10 ^b^	0.02 ± 0.01 ^d^	0.73 ± 0.12 ^b^	n.d.	n.d.	n.d.	n.d.	n.d.	n.d.	n.d.	n.d.	n.d.	n.d.	n.d.	n.d.
2	50	17.5	42.91	3.30	30.78	20.03 ± 2.09 ^e^	6.17 ± 0.87 ^d^	3.86 ± 0.53 ^d^	1.69 ± 0.27 ^e^	6.25 ± 0.41 ^e^	3.59 ± 0.32 ^e^	0.99 ± 0.20 ^a^	0.96 ± 0.12 ^e^	1.66 ± 0.21 ^b^	0.04 ± 0.02 ^d^	0.78 ± 0.09 ^b^	n.d.	n.d.	n.d.	n.d.	n.d.	n.d.	n.d.	n.d.	n.d.	n.d.	n.d.	n.d.
3	50	30	41.41	3.50	29.36	24.92 ± 1.87 ^e^	9.12 ± 1.02 ^e^	4.23 ± 0.36 ^d^	1.78 ± 0.21 ^e^	6.08 ± 0.36 ^e^	6.01 ± 0.45 ^b^	1.24 ± 0.32 ^a^	0.58 ± 0.16 ^d^	1.66 ± 0.23 ^b^	0.03 ± 0.01 ^d^	0.71 ± 0.13 ^b^	n.d.	n.d.	n.d.	n.d.	n.d.	n.d.	n.d.	n.d.	n.d.	n.d.	n.d.	n.d.
4	115	5	41.92	12.35	37.98	22.63 ± 1.79 ^e^	12.35 ± 0.90 ^b^	4.77 ± 0.19 ^ed^	2.17 ± 0.34 ^e^	6.30 ± 0.75 ^e^	9.37 ± 0.67 ^a^	1.18 ± 0.29 ^a^	1.66 ± 0.15 ^b^	0.98 ± 0.30 ^e^	0.33 ± 0.05 ^e^	1.03 ± 0.10 ^b^	8.79 ± 0.57 ^a^	n.d.	3.61 ± 0.24 ^a^	n.d.	3.66 ± 0.23 ^a^	n.d.	1.52 ± 0.10 ^b^	n.d.	n.d.	n.d.	n.d.	n.d.
5	115	17.5	43.41	13.01	37.80	30.41 ± 1.90 ^b^	13.85 ± 1.21 ^b^	4.12 ± 0.56 ^d^	1.53 ± 0.33 ^e^	7.96 ± 0.54 ^b^	10.33 ± 1.03 ^a^	0.92 ± 0.35 ^a^	0.88 ± 0.20 ^ed^	1.26 ± 0.18 ^b e^	0.40 ± 0.04 ^e^	0.91 ± 0.30 ^b^	7.84 ± 0.60 ^b^	0.21 ± 0.09 ^a^	2.87 ± 0.18 ^b^	0.09 ± 0.03 ^b^	2.96 ± 0.19 ^b^	0.06 ± 0.04 ^a^	1.84 ± 0.21 ^b^	0.05 ± 0.01 ^a^	0.09 ± 0.02 ^e^	n.d.	0.07 ± 0.01 ^e^	n.d.
6	115	17.5	45.22	13.37	38.08	32.72 ± 1.07 ^b^	12.66 ± 1.03 ^b^	4.65 ± 0.25 ^d^	2.05 ± 0.18 ^e^	7.84 ± 0.67 ^b^	9.38 ± 0.78 ^a^	1.11 ± 0.23 ^a^	1.16 ± 0.08 ^e^	1.54 ± 0.27 ^b^	0.33 ± 0.04 ^e^	0.96 ± 0.21 ^b^	7.35 ± 0.78 ^b^	0.34 ± 0.10 ^a^	2.52 ± 0.30 ^b^	0.15 ± 0.02 ^a^	2.94 ± 0.27 ^b^	0.14 ± 0.03 ^a^	1.65 ± 0.18 ^b^	0.05 ± 0.02 ª	0.12 ± 0.03 ^e^	n.d.	0.12 ± 0.03 ^a b^	n.d.
7	115	17.5	44.85	13.52	37.34	31.96 ± 1.68 ^b^	13.06 ± 0.79 ^b^	6.48 ± 0.37 ^e^	2.46 ± 0.51 ^e^	8.80 ± 1.10 ^b^	10.42 ± 1.12 ^a^	1.04 ± 0.10 ^a^	1.11 ± 0.10 ^e^	1.45 ± 0.16 ^b^	0.23 ± 0.05 ^e^	0.90 ± 0.07 ^b^	9.56 ± 0.61 ^a^	0.33 ± 0.07 ^a^	2.92 ± 0.15 ^b^	0.15 ± 0.01 ^a^	3.67 ± 0.32 ^a^	0.13 ± 0.02 ^a^	2.59 ± 0.09 ^a^	0.06 ± 0.02 ^a^	0.30 ± 0.05 ^a^	n.d.	0.08 ± 0.02 ^b^	n.d.
8	115	30	44.78	12.52	39.11	29.63 ± 2.83 ^b^	12.74 ± 0.87 ^b^	5.16 ± 0.69 ^e^	3.36 ± 0.42 ^b^	8.65 ± 0.98 ^b^	8.96 ± 0.98 ^a^	1.01 ± 0.12 ^a^	1.08 ± 0.25 ^e^	1.06 ± 0.09 ^b e^	0.36 ± 0.03 ^e^	0.94 ± 0.13 ^b^	10.13 ± 0.99 ^a^	0.27 ± 0.12 ^a^	2.99 ± 0.20 ^b^	0.11 ± 0.03 ^b^	3.81 ± 0.39 ^a^	0.10 ± 0.04 ^a^	2.98 ± 0.32 ^a^	0.06 ± 0.01 ^a^	0.18 ± 0.04 ^b^	n.d.	0.16 ± 0.04 ^a^	n.d.
9	180	5	63.66	20.22	45.00	36.30 ± 1.08 ^a^	15.76 ± 1.76 ^b^	12.67 ± 0.89 ^b^	3.52 ± 0.38 ^b^	11.62 ± 1.01 ^a^	8.70 ± 0.56 ^a^	0.95 ± 0.09 ^a^	2.31 ± 0.31 ^a^	2.44 ± 0.21 ^a^	1.18 ± 0.12 ^a^	1.41 ± 0.09 ^a b^	6.45 ± 0.60 ^b e^	0.30 ± 0.08 ^a^	1.70 ± 0.16 ^e^	0.10 ± 0.01 ^b^	2.69 ± 0.25 ^b^	0.12 ± 0.02 ^a^	1.88 ± 0.15 ^b^	0.08 ± 0.03 ^a^	0.11 ± 0.02 ^e^	n.d.	0.07 ± 0.02 ^b^	n.d.
10	180	17.5	65.66	22.72	46.35	34.69 ± 1.12 ^a^	21.65 ± 1.24 ^a^	21.10 ± 1.09 ^a^	4.53 ± 0.13 ^a^	12.46 ± 0.79 ^a^	9.94 ± 0.78 ^a^	1.19 ± 0.23 ^a^	2.15 ± 0.19 ^a^	1.93 ± 0.30 ^a b^	1.04 ± 0.08 ^a^	1.74 ± 0.14 ^a^	5.90 ± 0.59 ^e^	0.29 ± 0.08 ^a^	1.76 ± 0.12 ^e^	0.09 ± 0.02 ^b^	2.23 ± 0.18 ^e^	0.13 ± 0.03 ^a^	1.69 ± 0.26 ^b^	0.06 ± 0.02 ^a^	0.17 ± 0.02 ^b^	n.d.	0.05 ± 0.01 ^e^	n.d.
11	180	30	69.27	22.88	49.47	35.11 ± 1.45 ^a^	20.74 ± 1.45 ^a^	22.35 ± 1.74 ^a^	4.49 ± 0.25 ^a^	12.55 ± 0.68 ^a^	10.24 ± 0.71 ^a^	1.17 ± 0.19 ^a^	1.22 ± 0.17 ^e^	2.58 ± 0.28 ^a^	0.78 ± 0.07 ^b^	1.62 ± 0.15 ^a^	6.03 ± 0.63 ^e^	0.27 ± 0.12 ^a^	1.84 ± 0.19 ^e^	0.09 ± 0.03 ^b^	2.34 ± 0.19 ^e^	0.11 ± 0.04 ^a^	1.63 ± 0.19 ^b^	0.07 ± 0.02 ^a^	0.13 ± 0.03 ^e^	n.d.	0.09 ± 0.03 ^ab^	n.d.

n.d. not detect. Different letters denote significant differences (*p* ≤ 0.05) between different truffle species (*p* ≤ 0.05).

**Table 4 foods-11-00298-t004:** Cellular antioxidant activity of PLE extracts obtained after 30 min extraction at 180 °C from both truffles using water (W), ethanol (E) and water:ethanol mixture (1:1) (E:W). ^A,B^ Different letters denote significant differences (*p* ≤ 0.05) between different extraction solvents for the same truffle specie ^a,b^.

Truffle Species	Extraction Solvent	EC_50_ (µg/mL)
*T. claveryi*	W	402.96 ± 5.91 ^A,a^
	E:W (1:1)	481.73 ± 9.80 ^B,b^
	E	795.58 ± 16.81 ^C,b^
*T. aestivum*	W	565.95 ± 8.77 ^A,b^
	E:W (1:1)	364.73 ± 6.94 ^A,a^
	E	578.71 ± 10.25 ^B,a^

Different letters denote significant differences (*p* ≤ 0.05) between different truffle species for the same solvent.

**Table 5 foods-11-00298-t005:** Glucosidase and amylase inhibitory activities (IC_50_) of PLE extracts obtained after 30 min extraction at 180 °C from both truffles ^a–f^.

Truffle Species	Extraction Solvent	α-Amylase IC_50_ (mG/mL)	α-Glucosidase IC_50_ (mG/mL)
*T. claveryi*	W	66.7 ± 2.59 ^d^	202.62 ± 3.84 ^f^
	E:W (1:1)	80.73 ± 3.61 ^e^	1.97 ± 0.64 ^c^
	E	195.52 ± 5.74 ^f^	0.01 ± 0.00 ^a^
*T. aestivum*	W	9.44 ± 2.64 ^b^	52.91 ± 2.99 ^e^
	E:W (1:1)	63.24 ± 1.98 ^d^	49.94 ± 5.11 ^e^
	E	52.32 ± 2.36 ^c^	7.94 ± 0.86 ^d^
Arcabose (1 mg/mL)		0.67 ± 0.03 ^a^	0.83 ± 0.05 ^b^

Different letters denote significant differences (*p* ≤ 0.05) for the same enzyme.

## Data Availability

Not applicable.

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
