# Peer review of "Application of Pressurized Liquid Extractions to Obtain Bioactive Compounds from Tuber aestivum and Terfezia claveryi"

_foods, 2022, doi:10.3390/foods11030298_

Round 1

Reviewer 1 Report

The research (experimental design) has a lot of experimental work, but it may be that the best operating conditions for optimization have not been established. The authors should make some changes referents to optimization to validate the results. 

CHECK LIST Manuscript Number:
Pressurized liquid extractions using RSM (response surface methodology) to obtain bioactive compounds from Tuber
aes- tivum and Terfezia claveryi.

The research (experimental design) has a lot of experimental work, but it may be that the best operating conditions for
optimization have not been established. The authors should make some changes referents to optimization to validate the
results.

Example:

In methodology, a 3-level design (3 2) is mentioned, with 2 factors, which does not make it clear, since it also
mentions the combination of solvents as variables. Clarify point 2.3 of the methodology that the solvent study is not part
of the factorial design, since this qualitative / fixed variable is carried out for all design runs. Then it is optimized
considering the best solvent resulting from each response variable. And in discussion (3.2) also mentions it as independent
factors, but it is not part of the factorial design, clarify because it confuses.

More data from the optimization, such as the ANOVA, equation, the coefficient of determination (r2), are expected
to support the validity of the model. It seems to be that it was calculated according to the observation of data, since
extreme conditions are considered optimal: what would have happened to 35 min? 200 ° C... they don't discuss it in the
document either.

Then, with all the response variables, a PCA analysis is carried out, it is not clear in methodology. And by results,
each of the types of solvents is considered.

And it seems that in the methodology it is not clear either, which antioxidant properties are carried out with the
different types of solvents and with the optimal conditions carried out in the previous stage.

Reviewer 2 Report

In the manuscript entitled "Pressurized liquid extractions using RSM (response surface methodology) to obtain bioactive compounds from Tuber aestivum and Terfezia claveryi", the authors studied the effect of pressurized liquid extraction on the extraction of bioactive compounds from Tuber aestivum and Terfezia claveryi. The manuscript is easy to follow, but there are some important concerns that I have mentioned in the individual sections.

Abstract: The quantitative data on the main results need to be mentioned in the abstract.

Introduction:

The authors need to give more information about the bioactives of the fungus. The introduction section needs to be supplemented with successful case studies on different equation techniques that can be used for extraction of bioactives from fungi. Why is PLE technique better than other extraction techniques?

Also the novelty of the research should be highlighted at the end of the introduction section.

Material and Methods:

Section 2.4. were measurements made with replicates and how were each result expressed?

Section 2.5. where were measurements made with replication and how were the individual results expressed?

Section 2.6. how were the individual results expressed?

Section 2.5. where were measurements made with replication and how were each result expressed?

Equation 1. please use the dot as the singular of multiplication.

Section 2.8. how was the normality of the data tested before ANOVA analysis?

Section 2.8. there is no information about the analysis of the data collected according to the experimental design.

Section 2.8. there is no information about the PCA analysis that was performed.

Results and Discussion:

In Table 2 and Table 3, the standard deviations or standard errors of the data should be reported.

RSM equations should also be presented to provide information on the individual influence of selected input variables

Line 269. how were the optimal conditions estimated?

Did the authors perform independent validation of the optimal conditions?

Why did the authors use PCA analysis if the data were analyzed using the RSM method?

In Table 4 and Table 5, the standard deviations or standard errors of the data should be reported.

Reviewer 3 Report

The manuscript described the application of PLE under various operating parameters (e.g., temperature, time and extraction solvent) for isolation of bioactive compounds and their ability to act against cell oxidation and as immunomodulators studied from Tuber aestivum and Terfezia claveryi.

General comments: it is interesting work and provide good information of the possibility of PLE to recover various BACs from Tuber aestivum and Terfezia claveryi.

The title should be reformulated so that it better represents everything that has been done in the paper. I am wondering why authors used RSM in the title?

Abstract: Results showing here was rather general, please provide the main numerical findings here.

Most of citations in the text are not correctly written as authors did not follow the instructions for authors (https://www.mdpi.com/journal/foods/instructions).

Page 2/18

Include the device and the regime used for freeze drying.

Page 3/18

Line 98 – what was the volume of ASE extraction cell?

Why authors used cellulose filters as it is well known that glass-fiber filters are recommended for aqueus extractions while cellulose filters are appropriate for extraction methods that use organic solvents? (http://tools.thermofisher.com/content/sfs/manuals/75689-Man-ASE-ASE350-Operators-Dec2011-DOC065220-04.pdf )

Line 102 -

It is clear from Smiderle et al. (2017) that the authors used a full factorial 32 experimental design with temperature: 50, 115, and 180 ℃ and extraction time: 5, 17.5, and 30 min. It is not clear why the authors did not include the number of cycles to investigate its influence, as it has already been established that the efficiency of PLE extraction is strongly influenced by all three extraction parameters (time, temperature, number of cycles).

It is also not clear how the authors were able to perform the extractions at 17.5 minutes, as the device does not have this option, but extractions can only be performed in full minutes?

Line 129 – what authors mean with “Truffle or PLE extracts”?

Page 5/18

Include statistic analysis for Table 1, Table 4 and Table 5, as from these results, credible conclusions can not be obtained.

Page 7/18 and 8/18

It is not clear what the results shown in Table 2 and Table 3 present, since they are not statistically analyzed, the values shown cannot be compared, nor can any valid conclusion be drawn from the results in this table. The results need to be statistically analyzed.

Page 9/18

It is necessary to increase the fonts in Figure 1 in order to increase the clarity.

Conclusions - I am wondering how the structure of extracts will not change under such a high temperature and long time (30 min/180 ℃).

Round 2

Reviewer 1 Report

The authors made the suggested changes. Methodology 2.3 was clarified and now agrees with discussion 3.2.
The authors changed the Title of the work, therefore, it is not necessary to add details to validate the optimization model. okay.
They also improved PCA methodology.

Author Response

Thank you very much for your words.

Reviewer 2 Report

The authors put an effort and answered the comments and suggestions. In my opinion, they have improved the manuscript. A small correction of the supplementary material - please use the decimal point instead of the decimal comma, and please use the dot as a sign of multiplication.

Author Response

Thank you very much for your words.

Supplementary material section has modified and dot is now uses as a sign of multiplication. We have corrected the mistake. 

Reviewer 3 Report

The manuscript has been improved according to the reviewer's comments.

Author Response

Thank you very much for your words.